# Characterization of SOD1-DT, a Divergent Long Non-Coding RNA in the Locus of the SOD1 Human Gene

**DOI:** 10.3390/cells12162058

**Published:** 2023-08-13

**Authors:** Marika Guerra, Lucia Meola, Serena Lattante, Amelia Conte, Mario Sabatelli, Claudio Sette, Camilla Bernardini

**Affiliations:** 1Department of Neuroscience, Section of Human Anatomy, Catholic University of the Sacred Hearth, 00168 Rome, Italy; lucia.meola1@unicatt.it (L.M.); claudio.sette@unicatt.it (C.S.); 2GSTeP-Organoids Research Core Facility, Fondazione Policlinico Universitario A. Gemelli IRCCS, 00168 Rome, Italy; 3Department of Biological and Environmental Sciences and Technologies, University of Salento, 73100 Lecce, Italy; serena.lattante@unisalento.it; 4Adult NEMO Clinical Center, Unit of Neurology, Department of Aging, Neurological, Orthopedic and Head-Neck Sciences, Fondazione Policlinico Universitario A. Gemelli IRCCS, 00168 Rome, Italy; amelia.conte@centrocliniconemo.it (A.C.); mario.sabatelli@unicatt.it (M.S.); 5Section of Neurology, Department of Neuroscience, Faculty of Medicine and Surgery, Università Cattolica del Sacro Cuore, 00168 Rome, Italy

**Keywords:** long non-coding RNA, SOD1-DT, ALS, SOD1, transcription

## Abstract

Researchers studying Amyotrophic Lateral Sclerosis (ALS) have made significant efforts to find a unique mechanism to explain the etiopathology of the different forms of the disease. However, despite several mutations associated with ALS having been discovered in recent years, the link between the mutated genes and its onset has not yet been fully elucidated. Among the genes associated with ALS, superoxide dismutase 1 (SOD1) was the first to be identified, but its role in the etiopathogenesis of the disease is still unclear. In recent years, research has been focused on the non-coding part of the genome to fully understand the mechanisms underlying gene regulation. Non-coding RNAs are conserved molecules and are not usually translated in protein. A total of 98% of the human genome is composed of non-protein coding sequences with roles in the transcriptional and post-transcriptional regulation of gene expression. In this study, we characterized a divergent nuclear lncRNA (SOD1-DT) transcribed in the antisense direction from the 5′ region of the SOD1 coding gene in both the SH-SY5Y cell line and fibroblasts derived from ALS patients. Interestingly, this lncRNA seems to regulate gene expression, since its inhibition leads to the upregulation of surrounding genes including SOD1. SOD1-DT represents a very complex molecule, displaying allelic and transcriptional variability concerning transposable elements (TEs) included in its sequence, widening the scenario of gene expression regulation in ALS disease.

## 1. Introduction

Long non-coding RNAs (lncRNAs) are transcripts longer than 200 nucleotides (nt) that rarely have protein coding potential and usually have regulatory functions [1]. To date, the GENCODE consortium has annotated 10,000–50,000 lncRNA genes in the human genome [2]. Many of these RNAs are localized in the regions between protein-coding gene loci, while others overlap with protein-coding genes in both antisense and sense orientations. LncRNAs range from small single exon transcripts to large multi-exonic transcripts and can produce different isoforms through alternative splicing. Remarkably, 40% of these lncRNAs are specifically expressed in the brain and have been associated with the evolution of the central nervous system (CNS) [3]. Many lncRNAs display neuronal-specific expression, suggesting an important role in the neuronal diversification, typical of higher vertebrates [4]. Additionally, the spatially and temporally restricted expression patterns of many lncRNAs indicate tight regulation, suggesting that they may control the specification and function of individual neuronal subtypes [5].

While functional characterization of neuronal-enriched lncRNAs is still limited, some studies have implicated them as regulators of transcription through both epigenetic regulation of chromatin structure and interaction with transcription factors (TFs) [6]. Importantly, an increasing amount of lncRNAs has also been associated with neurodegenerative diseases. Examples include beta-secretase 1 (BACE1)-AS, whose expression is increased upon amyloid-beta 1–42 production in Alzheimer’s disease [7], and nuclear-enriched abundant transcript 1 (NEAT1), that is upregulated in different models of Parkinson’s disease interacting with the PTEN-induced kinase 1 (PINK1) protein [8]; Finally, several lncRNAs, such as DGCR5 and HAR1, are dysregulated in Huntington’s disease [9].

Among the neurodegenerative disorders, ALS is a heterogeneous disease, characterized by the degeneration of motor neurons. The main symptoms of ALS comprise motor dysfunction, muscle weakness, spasticity, and dysphagia, with some patients developing cognitive and/or behavioral impairments. In recent decades, progress has been made in identifying the genetic causes of ALS, and in particular, the coding portion of the genome has received much attention due to idea that dysfunctional proteins could lead to the death of motor neurons.

The first coding gene directly associated with ALS is that encoding superoxide dismutase 1 (SOD1); SOD1 mutations occur in approximately 18.9% of familial ALS (fALS) cases and in 1.2% of sporadic ALS (sALS) cases [10]. Notably, the link between ALS and SOD1 dysfunction has not yet been explained and questions still remain regarding the regulation of SOD1 transcription in healthy and ALS cells.

However, during the last few years, mounting evidence has pointed to altered RNA metabolism as a hallmark of ALS pathogenesis [11]. This notion is supported by the discovery of pathogenic variants in genes encoding for RNA binding proteins (RBPs), such as such as TAR DNA-binding protein 43 (TDP-43), fused in sarcoma (FUS), ataxin-2 (ATXN2), TATA-box-binding protein-associated factor 15 (TAF15), and heterogeneous nuclear ribonucleoprotein A1 (HNRNPA1) and A2/B1 [12].

Although mutations in several genes have been associated with ALS disease [13], a common pathological mechanism is still missing, and efforts have also been directed towards the non-coding part of the genome. One of the first lncRNAs associated with ALS was NEAT1_2, which has an important role in the formation of paraspeckle in motor neurons, in the early phase of the disease. This lncRNA prevents the wrong responses to a stressful condition; otherwise, cells switch to an apoptotic fate to repress inflammation caused by cellular stress [14].

Herein, we have focused on a recently identified lncRNA [15,16], annotated as superoxide dismutase 1 divergent transcript (SOD1-DT), which is near the SOD1 gene and transcribed in the opposite direction. Although SOD1-DT has already been annotated in previous analyses based on reported signals of transcriptional activity and epigenetic marks of active genes near its transcription start site (TSS), no information has yet become available regarding the expression and function of this lncRNA in the context of ALS. From this perspective, we investigated and characterized SOD1-DT, both at the genetic and transcriptional levels, in a neuron-like context and in a cohort of ALS patients. Our study indicates that SOD1-DT is expressed in neuronal cells and affects the expression of SOD1 and other flanking genes, suggesting the functional relevance of this previously uncharacterized lncRNA.

## 2. Materials and Methods

### 2.1. In Silico Analysis

In silico analysis was performed by using the UCSC Genome Browser [17] (Human Assembly GRCh38/hg38) with the following tracks: RepeatMasker, Gene Hancer, and Registry of Candidate cis-Regulatory Elements (cCREs). The analysis of TFs binding to the selected promoter-like element was performed with the SCREEN V2 tool, which is a web interface for searching and visualizing the Registry of Candidate cis-Regulatory Elements (cCREs) derived from ENCODE data. TFBIND software (https://tfbind.hgc.jp, accessed on the 20 December 2021) was used for searching transcription factor binding sites; this software calculates the similarity (0.0–1.0) between a registered sequence for the transcription factor binding sites and the input sequence. 

The Alternative Splice Site Predictor (ASSP) [18] predicts putative alternative exon isoform, cryptic, and constitutive splice sites of internal (coding) exons; this tool was used to predict alternative splice sites in the SOD1-DT sequence. The web application for the prediction of alternative splice sites is available at http://es.embnet.org/~mwang/assp.html (accessed on the 13 March 2022).

### 2.2. Cell Culture, Cell Fractionation, and SH-SY5Y Differentiation

Commercial lines of SH-SY5Y (ATCC^®^ CRL2266™, Manassas, VA, USA) cells were cultured in Dulbecco′s Modified Eagle′s Medium (DMEM) supplemented with 10% fetal bovine serum (FBS), 10,000 U/mL penicillin, 10,000 U/mL streptomycin, Zellshield reagent (Minerva Biolabs, Berlin, Germany), and 10 mM non-essential amino acids (NEAA) (Gibco, ThermoFisher, Waltham, MA, USA). To induce the neural differentiation process in SH-SY5Y, cells were treated with all-trans retinoic acid (RA) (Gibco, ThermoFisher, Waltham, MA, USA). Cells were seeded in a 100 mm plate at a 75 × 10^4^ cells/cm^2^ density and cultured in DMEM supplemented with 3% FBS and 10 µM RA. Cells were collected at three different time points: 0, 3, and 6 days, for differentiation control.

Primary fibroblasts derived from both ALS patients carrying eight SOD1 mutations (Appendix A) and from two healthy controls were obtained as previously described [19]. Skin biopsies were performed after obtaining informed consent. The study was approved by our institution’s Ethics Committee (Protocol nr. A.133/CE/2013). Fibroblasts were cultured in DMEM, supplemented with 20% FBS, 10,000 U/mL penicillin, 10,000 U/mL streptomycin, Zellshield reagent, and 10 mM NEAA. All cell lines were incubated at 37 °C under 5% CO_2_ and a 95% air-humified atmosphere. 

Cytosol/nucleoplasm/chromatin fractionation was performed as previously described [20]. 

### 2.3. Immunofluorescence Assay and Neurite Quantification

Cells were fixed with 4% paraformaldehyde (PFA) in Phosphate Buffered Saline (PBS) for 20 min, washed in PBS, permeabilized and blocked using 1% Bovine Serum Albumin (BSA) and 0.1% Triton in PBS. Cells were then incubated with primary antibody for βIII-tubulin (Sigma-Aldrich, Merck, St. Louis, MO, USA) (1:200) for 1 h at room temperature (RT). They were successively washed with 1% BSA in PBS, incubated with Alexa Fluor^®^ 488 secondary antibody (Invitrogen, Thermo Fisher, Waltham, NA, USA) (1:200) for 1 h at RT, and washed again with 0,1% BSA in PBS. Finally, cells were incubated for 5 min with Hoechst (Invitrogen, Thermo Fisher, Waltham, MA, USA) (1:1000) for nuclei staining. Images were acquired with Zeiss Axio Observer fluorescence microscope (Zeiss, Oberkochen, Baden-Württemberg, Germany).

Morphometric analysis of SH-SY5Y cells was performed on βIII-tubulin immunofluorescence images through ImageJ software (National Institutes of Health, Bethesda, MD, USA), version 1.53t. The NeuronJ plugin was applied for semi-automated neurite tracing. Neurite count and total length values (µm) were expressed as a ratio of the number of cells per image, which was assessed by the Cell Counter plugin.

### 2.4. DNA Collection, RNA Isolation, PCR, qPCR, and Sanger Sequencing

Genomic DNA (gDNA) was extracted from the cells and from the peripheral blood of ALS patients by using a Wizard Genomic DNA Purification Kit (Promega, Madison, WI, USA). Sanger Sequencing was used to analyze SOD1 coding regions. For this analysis, we employed 23 fALS and 49 sALS patients; precisely, 18 fALS and 21 sALS patients, altogether carrying 20 different SOD1 mutations, while the remaining patients had any detected mutation (Appendix A). Genotype and allele frequencies were calculated by dividing the number of individuals with each genotype/allele by the total sample population, expressed as percentages. 

Total RNA was isolated using the Trizol reagent method (Invitrogen) according to the manufacturers’ instructions. Genomic DNA was digested by RNase-free Dnase (Roche, Basel, Switzerland). RNA purity and concentration were quantified using the NanoDrop 2000 UV spectrophotometer (ThermoFisher, Waltham, MA, USA). A total of 1 µg of total RNA was reverse-transcribed with random primers (Sigma-Aldrich, Merck, St. Louis, MO, USA) using M-MLV reverse transcriptase (Promega, Madison, WI, USA). CDNA was used as template for PCR and qPCR analyses. RPL34 was used as a housekeeping gene. Reaction products were analyzed on 2,5% agarose gel, stained with ethidium bromide (Sigma-Aldrich, Merck, St. Louis, MO, USA).

CDNA was purified from agarose gel using a QIAquick Gel Extraction Kit (QIAGEN, Hilden, Germany) and processed for Sanger sequencing with the Mix2Seq Kit (Eurofins, Nantes, Bruxelles), according to the manufacturer’s protocol. The quality of sequencing results was analyzed with the Snap Gene Viewer Software™ version 7.0 and aligned with the NCBI-BLAST™ tool to confirm the identity of each transcript.

Quantitative PCR reactions were performed using the LightCycler 480 System with SYBR Green I Master Mix (Roche, Basel, Switzerland) following the manufacturer’s instructions. The 2-ΔCt method was applied to calculate differences in gene expression using the RPL34 gene for data normalization. The primers used in this study are listed in Appendix A.

### 2.5. Cell Transfection for Silencing and Overexpression

SH-SY5Y cells were seeded in a 6-well plate at a density of 4 × 10^5^ cells/cm^2^. Transfection was performed in DMEM with 10% FBS and 10 mM NEAA, without antibiotic–antimycotic solution. This was carried out in the presence of Opti-MEM^®^ medium (Gibco ThermoFisher, Waltham, MA, USA), by using the Lipofectamine^®^ RNAiMAX transfection reagent (Invitrogen, ThermoFisher, Waltham, MA, USA) according to the manufacturer’s instructions. Incubation of SH-SY5Y cells with siRNAs occurred for 24 h. Three different siRNAs (Sigma-Aldrich, Merck, St. Louis, MO, USA) at a final concentration of 50 nM were tested to specifically match the exons of SOD1-DT and to exclude off-target effects. The sequences of siRNAs used for this study were the following: 5′-AGTACGCGAAATTGGCAAA-3′ (Ex1), 5′-GAGAAAAGAATGTGTTGAA-3′ (Ex2), and 5′-TAGCTGGTGTGTCCGGAATT-3′ (Ex3).

The overexpression assay was performed as described above by using the Lipofectamine^®^ 3000 transfection reagent (Invitrogen, ThermoFisher, Waltham, MA, USA) according to the manufacturer’s instructions. PCMV6-AC-GFP and YY1-pCMV6-AC-GFP (OriGene, Herford, Germany) plasmids were used.

### 2.6. Western Blot Analysis

Total proteins were extracted from cells using RIPA buffer (50 mM Tris pH 7.4; 1% NP-40; 0.5% Na deoxycholate; 0.1% SDS; 150 mM NaCl; 1 mM EDTA) supplemented with a protease inhibitor cocktail (Sigma-Aldrich, Merck, St. Louis, MO, USA), 0.5 mM Na3VO4, and 1 mM DTT. Lysates were incubated on ice for 20 min, briefly sonicated, and centrifuged for 10 min at 13,000 rpm, 4 °C. Protein extracts were quantified via the Bradford assay, resolved at a final concentration of 20 μg by electrophoresis through 8–12% SDS-PAGE, and blotted on a PVDF membrane (Amersham, Buckinghamshire, UK). Blots were firstly incubated with a blocking solution (5% non-fat dry milk in PBS) for 1 h at 25 °C, and then, with the following primary antibodies: mouse anti-HSP90 (Santa Cruz, Dallas, TX, USA), rabbit anti-Lamin B (Santa Cruz, Dallas, TX, USA), mouse anti-GAPDH (Santa Cruz, Dallas, TX, USA), mouse anti-H3 (Santa Cruz, Dallas, TX, USA), mouse anti-β-ACTIN (Santa Cruz, Dallas, TX, USA), and rabbit anti-YY1 (OriGene, Herford, Germany). Antibodies were used in a 1:1,000 dilution in a 5% BSA in PBS solution and incubated overnight at 4 °C. Blots were washed and incubated with primary and secondary antibodies. Anti-rabbit and anti-mouse (GE HealthCare, Chicago, IL, USA) HRP-linked secondary antibodies were used at a 1:10,000 dilution in 5% non-fat dry milk in PBS, and the ECL signal was developed using the Clarity Western ECL Blotting Substrate (Biorad, Hercules, CA, USA).

### 2.7. Quantification and Statistical Analysis

Quantitative data were expressed as the mean ± standard deviation (SD), as indicated in the figure legends (* *p* value ≤ 0.05; ** *p* value ≤ 0.01; *** *p* value ≤ 0.001; **** *p* value ≤ 0.0001). Unpaired *t*-test and one-way analysis of variance (ANOVA) followed by Bonferroni’s multiple comparison post-test were performed using Prism 6 software (GraphPad Software version 7.0). 

## 3. Results

### 3.1. SOD1-DT Is an Uncharacterized Divergent lncRNA in the SOD1 Gene Locus

The inspection of the SOD1 locus, using the Genome Browser web tool, highlighted the existence of a divergent lncRNA (SOD1-DT). It extended in the antisense direction from the 5′ region of the SOD1 gene, from which, it is only 123 bp distant. SOD1-DT includes three exons and the analysis of the Human Expressed Sequence Tags (ESTs) track showed that SOD1-DT can generate different transcripts, either including or not including a transposable element (TE) in exon 2. In fact, by applying the RepeatMasker track, which screens for interspersed repeats and low-complexity DNA sequences, we found that the T1 and T2 transcripts (Figure 1A) differed only in terms of the presence of a LINE (L3) element belonging to the CR1 family. Moreover, a DNA transposon belonging to the hAT-Tip100 family was embedded within the exon 2 of both SOD1-DT transcripts (Figure 1A), while a LTR12C repeat belonging to the ERV1 endogenous retrovirus family was in exon 3 (Figure 1B).

A 2886 bp long promoter/enhancer element (GH21J031658) spanning from the SOD1-DT to the SOD1 gene, and a promoter-like element (EH38E2136874), were also identified by using the Gene Hancer and ENCODE Registry of Candidate cis-Regulatory Elements (cCREs) tracks in the USCC Genome Browser (Figure 1B). Combining the lists of SOD1-DT putative gene interactors/targets acquired through these tools, we identified SOD1, TIAM1, HUNK, CCT8, FBXW11P1, SCAF4, CFAP298, MIS18A, and URB1 genes as being potentially linked to SOD1-DT. These genes span a 2.5 Mb genomic region flanking the SOD1 gene, suggesting that SOD1-DT may act at considerable distances.

### 3.2. SOD1-DT Gene Analysis in SH-SY5Y Cells Reveals the Existence of Allelic Variability and Different Transcriptional Variants

To test whether SOD1-DT is expressed in neuronal cells, we designed PCR primers spanning from exon 1 to exon 3 to amplify the predicted T1 and T2 transcripts. Based on the mRNA sequence reported in the NCBI database (AP000253.1), we expected two products. However, RT-PCR analyses of RNA isolated from SH-SY5Y neuroblastoma cells revealed the existence of four different transcripts. In addition to the expected T1 (518 bp) and T2 (386 bp) products, the SOD1-DT gene expressed two other alternative transcripts (467 and 335 bp) (Figure 2A).

As was confirmed by Sanger Sequencing, T1 and T2 share the same sequence except for the L3 LINE element in exon 2 (Figure 2B). Furthermore, we found that the unannotated ΔT1 and ΔT2 transcripts are identical to T1 and T2, respectively, but lack a TE belonging to the hAT-Tip100 family (Figure 2B). This last feature is due to an allelic variability given by this transposable element, as shown by the PCR analysis of genomic DNA from SH-SY5Y cells (Figure 2C).

Observing, in detail, the inclusion of L3 in SOD1-DT exon 2, we realized that the TE was not fully included in the mature transcript. In fact, the LINE starts at −7 bp before the exon 2 start point, and the AG acceptor site (AS1) is included in the LINE itself. Beyond the splice site, the LINE also included a polypyrimidine tract (PY) preceding the successive acceptor site (AS2), which is located between the two TEs (Figure 2D). An analysis of these splice sites was performed with ASSP. The results indicated that both the splice sites are constitutive acceptors and AS1, which corresponds to the transcript with the included LINE, showed the higher score (AS1 = 9.605; AS2 = 5.121) (Figure 2D).

### 3.3. SOD1-DT lncRNA Is Enriched in the Chromatin Sub-Cellular Fraction

Generally, lncRNAs have a specific sub-cellular distribution that is critical for their function. Some lncRNAs are enriched in the nucleus and are involved in regulating nuclear mechanisms such as transcription and RNA processing; others are enriched in the cytoplasm where they can impact on protein localization or modulate mRNA stability and translation; some others are equally distributed between the nucleus and the cytoplasm [1]. SOD1-DT transcript subcellular localization was assessed, performing a cytosol/nucleoplasm/chromatin fractionation in SH-SY5Y cells. Western blot analysis of these subcellular fractions confirmed enrichment of HSP90 and GAPDH in the cytosol, of histone H3 in the chromatin, and of LAMIN B in the nucleoplasm fraction (Figure 2E). Interestingly, RT-PCR analysis of SOD1-DT distribution revealed that this lncRNA is enriched in the chromatin fraction, compared to the cytosol and nucleoplasm fractions (Figure 2F). These results suggest that SOD1-DT could be a chromatin-associated lncRNA regulating gene expression via different mechanisms: modifying chromatin organization, acting at the transcriptional and/or post-transcriptional level, or being a structural scaffold of nuclear domains; these are all functions that can be ascribed to nuclear lncRNAs.

### 3.4. SOD1-DT Transcription Is Up-Regulated in Differentiated SH-SY5Y

LncRNAs can be differentially expressed across various stages of differentiation, indicating that they might be fine tuners of cell fate [21]. Since SH-SY5Y cells maintain their potential for neuronal differentiation under culture and can be differentiated by treatment with RA [22], we tested whether the expression of SOD1-DT varies after inducing the neuronal differentiation process. For this purpose, SH-SY5Y cells were treated with 10 µM RA for six days. RA-mediated differentiation of SH-SY5Y induced a change in the phenotype from an epithelial-like state to a more typical neuronal-like state with distinct neurite processes. To confirm the outcome of the differentiation process, the length of neurites was measured in undifferentiated cells (D0) and three and six days after RA treatment (D3 and D6). As expected, RA induced a significant increase in neurite processes and length. At the same time, the morphological features of neurons were checked by β-III tubulin immunostaining, which showed an increase in β-III-tubulin-positive cells from D0 to D6 (Appendix A).

The SOD1-DT expression level was analyzed in undifferentiated and differentiated SH-SY5Y cells by RT-PCR. A statistically significant increase was found for all SOD1-AS transcripts at D6 compared to undifferentiated cells, as shown by the densitometric analysis (Figure 3A).

SOD1 expression did not change in differentiated cells (Appendix A). Moreover, a smaller amplicon of around 110 bp was expressed in differentiated cells, but not in proliferating cells at D0. Sequencing analysis indicated that this isoform lacks the entire exon 2 and 145 bp at the 5′ of the exon 3, corresponding to a SINE element belonging to the Alus family (Figure 3B).

### 3.5. YY1 TF Overexpression Promotes SOD1-DT Gene Expression In Vitro

To explore the potential activity markers and TFs binding of the promoter region of SOD1-DT, additional in silico exploration was provided by the SCREEN (Search Candidate cis-Regulatory Elements) web tool. The analysis indicated that the promoter-like element (EH38E2136874) displayed activity signs in all registry biosamples; interestingly this 349 bp long element partially overlaps the SOD1 exon 1 (Figure 1B). By querying the SCREEN V2 database, we obtained a list of transcription factors with ChIP-seq peaks that intersected the selected cCRE. The results displayed different TFs targeting the regulatory element mentioned before (EH38E2136874) (Appendix A). Among them, the top ranked TFs were Ying Yang 1 (YY1), RNA Polymerase II Subunit A (POLR2A), CCCTC-Binding Factor (CTCF) and TATA-Box-Binding Protein-Associated Factor 1 (TAF1).

To define whether SOD1-DT could be upregulated by a specific transcription factor, we selected YY1 for successive experiments, because it is a ubiquitously distributed TF belonging to the GLI-Kruppel class of zinc finger protein, that is already known to regulate several lncRNAs [23].

From the further analysis of the regulatory element EH38E2136874 with the TFBIND tool, we noticed a YY1 binding site located at −29 bp before the annotated transcription start site (TSS) for SOD1-DT (Figure 4A). 

This binding site shows similarity (0.75) between a registered sequence for the transcription factor binding site and the input sequence (NNNCGGCCATCTTGNCTSNW ACGCGGCCCCTTGGCCCCGC). Given these indications, we decided to assess whether the overexpression of YY1 could vary the expression of SOD1-DT by using a GFP-tagged expression vector in SH-SY5Y cells. To verify the outcome of the transfection, the percentage of transfected cells was checked by using fluorescence microscope visualization and by Western Blot analysis with an antibody targeting YY1 (Appendix A). Analysis of SOD1-DT expression levels by RT-PCR showed increased expression of SOD1-DT in YY1-overexpressing cells (Figure 4B), indicating that this lncRNA can be upregulated in vitro following the overexpression of YY1. 

### 3.6. SOD1-DT Knockdown Modulates SOD1 and Other Gene Targets Expression In Vitro

An important step in lncRNAs characterization is to elucidate their involvement in cell and molecular biology, and one of the main roles of nuclear lncRNAs is the regulation of gene expression at the transcriptional level. Some lncRNA are trans acting, producing one or several transcripts affecting the genomic regions that are spatially distant from their site of production [24]. In contrast, others are cis acting, having an enhancer-like function for genes on the same chromosome [25]. To evaluate whether SOD1-DT could influence gene expression, its knock-down in SH-SY5Y cells was achieved through small interfering RNAs (siRNA), and three different siRNAs were designed to target SOD1-DT. An SiRNA targeting exon 1 was selected because it gave the best knockdown outcome (Figure 5A).

Since SOD1 was the closest gene to the SOD1-DT locus, we first assessed its mRNA expression levels by qPCR. The results indicated that following SOD-DT knockdown, SOD1 mRNA levels consequently increased (Figure 5B). Then, we evaluated the expression of other putative target genes, showing that a decrease in SOD1-DT resulted in a higher expression of TIAM1, HUNK, CCT8, and FBXW11P1 (Figure 5C). Otherwise, expression of the other potential target genes (SCAF4, CFAP298, MIS18A, and URB1) did not change (Appendix A). Taken together, these results suggest that SOD1-DT could regulate the expression of some of surrounding genes, also affecting genes at a distance greater than 2,5 Mb, such as CCT8 (Figure 5D).

### 3.7. SOD1-DT Expression in ALS Patients

The SOD1 gene is the second ALS-linked gene in terms of frequency, following C9ORF72 [26]. Although the association between SOD1 mutations and ALS is well known, it is not yet clear how alterations in the expression of SOD1 can affect the development of the disease, in both sALS and fALS [27]. Considering that changes in SOD1 mRNA levels have been associated with sALS, a molecular understanding of the processes involved in the regulation of SOD1 gene expression could clarify the mechanisms underlying the etiopathology of ALS disease.

As SOD1-DT knockdown induces a variation in the expression of the SOD1 gene, we began to investigate the potential relevance of SOD1-DT transcription in an ALS context by analyzing SOD1-DT expression in and skin fibroblasts derived from healthy controls and ALS patients. To this end, we examined SOD1-DT at both the genetic and transcriptional levels in eight ALS patients carrying different SOD1 mutations (Appendix A) and in two healthy controls. Genomic DNA analysis showed that the controls and most ALS patients (ALS1, ALS2, ALS3, ALS4, ALS6, ALS8) were homozygotes for the allele including the TE, while only ALS5 and ALS7 showed both the alleles. At the RNA level, ALS1, ALS2, ALS4, ALS8, and the two controls expressed only the transcript lacking the LINE element (386 bp); ALS6 expressed both transcript variants (with and without the LINE); finally, SOD1-DT was not expressed in ALS3. ALS5 and ALS 7 were heterozygous, showing both the alleles, with the former expressing all the transcripts and the latter only the transcripts without the LINE element (Figure 6A).

To define the frequency of the two SOD1-DT allelic variants (with/without the hAT-Tip100 TE), we analyzed the SOD1-DT allelic variability by PCR in genomic DNA from blood samples in a larger cohort of individuals. This included 63 age-matched healthy controls, and 23 fALS and 49 sALS patients. In more detail, 18 fALS and 21 sALS patients were SOD1-mutated, while none of the others presented any known ALS-related mutation. Globally, there are twenty different known SOD1 mutations (Appendix A). As expected, we observed three genotypes (Figure 6B): the homozygous genotype with the hAT-Tip100 TE (T+/+), the heterozygous genotype (T+/−) and the homozygous genotype lacking the hAT-Tip100 TE (T−/−). The T+/+ genotype occurred most frequently in the general population, with a rate of 68.25% in the controls and 80.56% in ALS patients, while the T−/− genotype occurred less frequently (1.59% in controls and 2.78% in ALS patients) (Figure 6C). Consequently, allelic frequency analysis showed that the T+ allele was predominant in both healthy individuals (83.33%) and patients with ALS (88.89%) (Figure 6D). We did not observe any significant difference between healthy individuals and ALS patients, from the analysis of 135 DNA samples (63 and 72 from the controls and ALS patients, respectively).

## 4. Discussion

During the present study we characterized SOD1-DT, a divergent lncRNA that is transcribed on the minus strand starting from the SOD1 gene promoter. LncRNAs contribute to many regulatory pathways, including chromatin organization, transcriptional regulation, and post-transcriptional and post-translational processing [1]. Most are developmentally regulated, abundantly expressed in the CNS, and change their expression during the differentiation process [4]. Divergent lncRNAs are transcribed from the opposite strand and comprise ∼20% of the total lncRNAs in mammalian genomes [28]. Although no clear function for this class of lncRNA has been identified, an interesting hypothesis is that divergent gene organization may allow lncRNA transcripts to regulate their neighboring coding genes [29,30].

SOD1-DT was previously investigated as a possible antisense transcription locus associated with hereditary neurodegenerative diseases [16]; furthermore, it has been found in plasma from patients with hepatitis B virus (HBV) infection, indicating a possible role as a plasmatic biomarker in HBV infection [15].

In our study, we characterized the structure of SOD1-DT in proliferating and differentiating the SH-SY5Y cell line, and in fibroblasts deriving from ALS patients; furthermore, we explored the potential connection with the target genes that were previously selected through an in silico analysis.

As detailed in the Results Section, the genetic variability of this lncRNA depends on two TEs annotated in the second exon; we identified two SOD1-DT alleles, either including or not including the DNA transposon (hAT-Tip100 TE), and each of them can produce two transcripts, either including or not including a LINE element (L3).

Since all these alternative isoforms are upregulated during the differentiation of neuroblastoma cells, we hypothesized that this modulation could reflect a functional role for SOD1-DT in a neural context. LncRNAs can drive neuronal differentiation through different pathways such as TFs recruitment to promoters [31], enhancing the expression of neuronal genes [32], modifying the balance of excitatory/inhibitory neurons [33], and controlling the balance between self-renewal and neuronal differentiation interacting with splicing regulators [34]. 

During our study, we assessed that SOD1-DT is enriched in the chromatin, compared to the cytoplasm and nucleoplasm fractions. This feature led us to direct our attention to those functions that can be circumscribed to the nucleus such as in cis gene expression regulation. We reasoned that this lncRNA could exert its function in different ways, such as the modification of chromatin organization, transcriptional and/or post-transcriptional gene regulation, or being a structural scaffold of nuclear domains, which are all functions that can be ascribed to chromatin-associated lncRNAs.

Working in this direction, we found that SOD1-DT modulates the gene expression of several neighboring genes in neuroblastoma cells. In fact, following its knockdown, we observed an increasing level in five (SOD1, TIAM1, HUNK, CCT8, and FBXW11P1) out of eight selected genes. The upregulation of SOD1 indicated that the divergent lncRNA associated with the SOD1 locus effectively affects the expression of the coding gene; in addition, interestingly, we also found TIAM1 to be an interactor of the NMDA receptor involved in dendritic spine morphology. TIAM1 could modify the stoichiometry of these receptors, consequently affecting the functions of neurons, and eventually leading to neurodegenerative disorders [35]. The other upregulated genes (HUNK, CCT8, and FBXW11P1) have not been described yet or directly associated with any disease, but are regulated after SOD1-DT knockdown, maybe for their proximity to its gene locus.

Overall, these data indicate that SOD1-DT knockdown upregulates the expression of several genes in a spatially defined manner. These results, together with SOD1-DT localization, suggest its role in the regulation of gene expression through chromatin structure modulation.

To further expand upon this idea, we also searched for TF binding sites in the SOD1-DT sequence, and we found an enrichment for CTCF and YY1 binding motifs. While CTCF is usually a mark of the boundaries of functionally connected regions that determine chromatin three-dimensional organization, YY1 is a ubiquitous TF that can activate or repress individual promoters. In mammals, YY1 regulates the expression of many lncRNAs [36,37]. For this reason, we focused our analysis on this TF, and we found that the overexpression of YY1 was able to increase SOD1-DT expression, suggesting that their interaction could be necessary for SOD1-DT function in gene expression regulation.

Among the analyzed target genes, SOD1 upregulation is noteworthy, since it is one of the genes whose mutations has been associated with familial form of ALS and its role in the disease is not yet fully understood [38]. To explore SOD1-DT in ALS disease, we analyzed its allelic variability in different DNA samples from both fALS and sALS patients. Although we did not observe a statistically significant difference in allelic frequency between the two ALS subgroups compared to the controls, we noticed that the allele including the DNA transposon in exon 2 was augmented progressively in sALS and fALS patients, in comparison to the control group. The functional significance of this data it is still not entirely clear, although allelic variation is a major driver of heritable phenotypic variation and a study in humans suggested that ∼10% of the ATGs of ORFs have a possible transposon origin [39].

Transposon insertions within lncRNAs is quite frequent, and the latest studies have reported that TEs might be involved in lncRNA origin and diversification, being functionally active in the genomes [40,41]. Although these elements appear to have lost the ability to move, they play a role in species evolution by contributing to new regulatory and coding sequences through a recruitment called exaptation or molecular domestication [42]. The LINE element, however, was always included in the DNA sequence, so we supposed that the RNA transcripts, differing only in the presence of this TE, could be generated by alternative splicing events. In fact, we found two canonical splicing sequences that are strictly related to the presence of the LINE element. It is known that TEs can function as splice donors or splice acceptors [43] depending on the splicing events. In fact, as is described in the Results Section, L3 includes in its sequence, the Acceptor Site (AS1) AG, which will produce the transcript including L3 itself. Moreover, it embraces the polypyrimidine tract (CTCTTCTTTT), which is employed by the successive Acceptor Site (AS2) producing the transcript without L3. Interestingly, although AS1 shows a higher value than AS2, only two out of eight ALS patients (ALS5 and ALS6) expressed the LINE element in their transcripts. Given its structure, the presence of LINE in the transcripts is essential to give variability to the transcripts of SOD1-DT, representing a compelling example of exaptation.

Altogether, our data are coherent with previous studies suggesting that divergent lncRNAs, or at least a subset of them, positively regulate the transcription of genes, acting in cis and participating in biological processes in which nearby genes are involved [28].

## 5. Conclusions

In conclusion, we have described SOD1-DT, a chromatin-associated lncRNA, upregulated during the neural differentiation process, producing different transcripts depending on the TEs included in its sequence. We determined that inhibition of this lncRNA leads to a change in the expression level of selected genes at a distance equal to 2.5 Mb and that YY1 can induce its expression level. Unfortunately, we did not find a direct correlation with ALS disease, but we are confident that this lncRNA deserves further investigation.

Hopefully, future studies of lncRNA/mRNA gene pairs and the lncRNA described here will provide new insights into the contributions of lncRNAs to the control of the cell state and the process of differentiation, possibly adding new perspective to the understanding of the etiopathology of neurodegenerative diseases.

## Figures and Tables

**Figure 1 cells-12-02058-f001:**
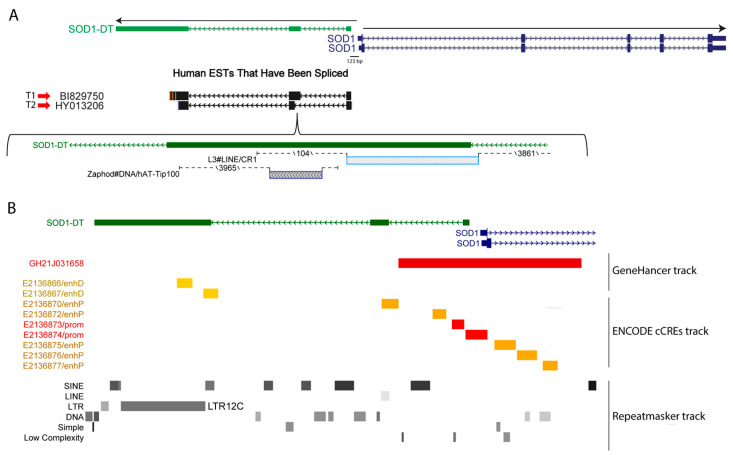
SOD1-DT features on the Genome Browser. (**A**) Basic gene annotation from GENCODE, version 34, for SOD1-DT (green) and SOD1 (blue) genes; the main human alternative transcripts analyzed in this work are highlighted. (**B**) Annotations for regulatory (Gene Hancer and ENCODE cCREs tracks) and repeated elements (Repeat masker track) found in SOD1-DT gene locus.

**Figure 2 cells-12-02058-f002:**
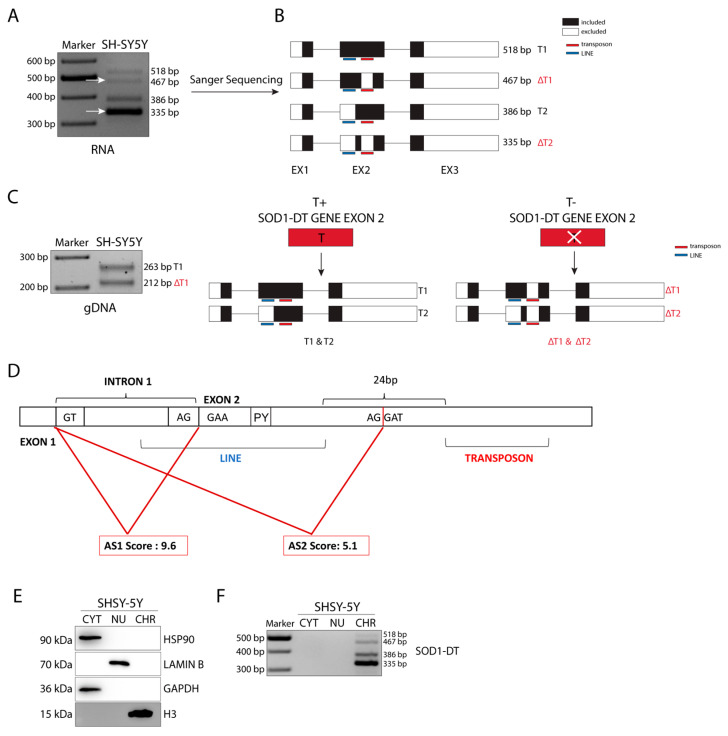
Characterization of SOD1-DT transcripts, allelic variability, and cellular localization in SH-SY5Y cells. (**A**) RT-PCR analysis of SOD1-DT and (**B**) schematic representation of Sanger sequencing results. Black, white, red, blue, and green bars show: the segment included in the transcripts, the segment excluded from the transcripts, transposon, LINE, and ALU elements in the sequence, respectively. (**C**) RT-PCR analysis on SOD1-DT gDNA and schematic representation of the two alleles generated from SOD1-DT. Black regions indicate the portion of transcripts covered for the amplification (**D**) SOD1-DT splice site analysis. (**E**) Western blot analysis on cell fractionation. (**F**) RT-PCR analysis of SOD1-DT on cytosol, nuclear, and chromatin fractionations. Expected products sizes are indicated in bp for RT-PCR and kDa for Western Blot analyses. The arrows indicate the two alternative transcripts.

**Figure 3 cells-12-02058-f003:**
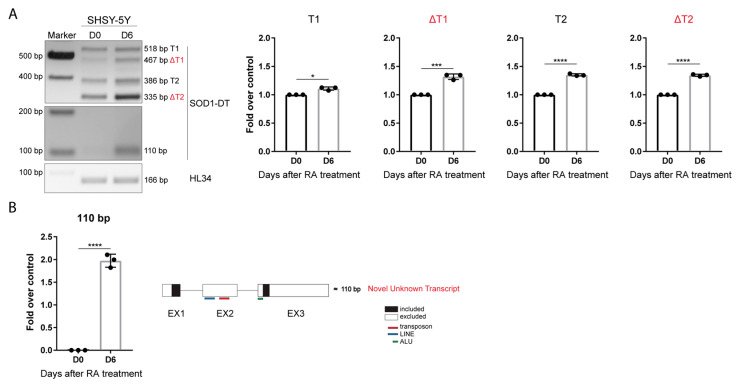
SOD1-DT in differentiated SH-SY5Y. (**A**) RT-PCR analysis of SOD1-DT transcripts in undifferentiated and differentiated SH-SY5Y, and relative densitometric analysis. Fold over control is shown. Data are expressed as: mean ± SD, * *p* value ≤ 0.05; *** *p* value ≤ 0.001; **** *p* value ≤ 0.0001; test = unpaired *t*-test. (**B**) A novel transcript of 110 bp is expressed in differentiated SH-SY5Y and schematic representation of Sanger sequencing results.

**Figure 4 cells-12-02058-f004:**
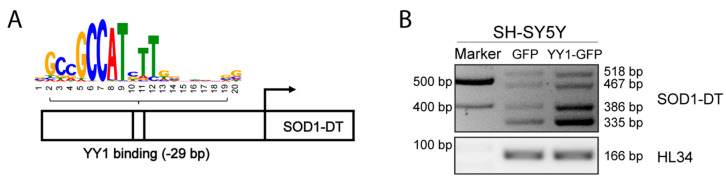
SOD1-DT is upregulated by YY1 TF in SH-SY5Y. (**A**) Predicted binding region of YY1 TF in SOD1-DT promoter. (**B**) RT-PCR analysis of SOD1-DT transcripts in YY1 overexpressed SH-SY5Y cells.

**Figure 5 cells-12-02058-f005:**
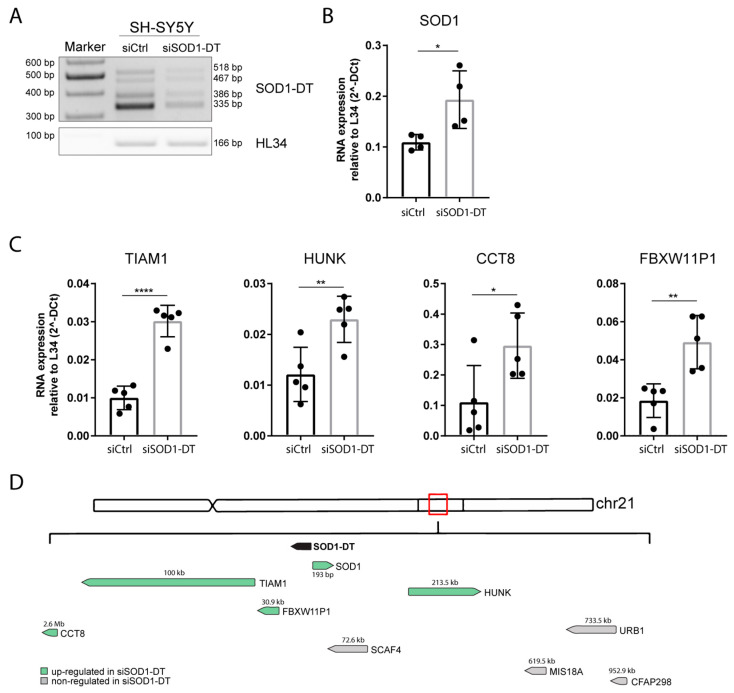
SOD1-DT silencing in SH-SY5Y and gene expression analysis of putative linked genes. (**A**) SOD1-DT RT-PCR (on the left) and (**B**) SOD1 qPCR (on the right) in silenced SH-SY5Y cells. (**C**) QPCR analysis of TIAM1, HUNK, CCT8, and FBXW11P1 genes in SOD1-DT-silenced cells compared to the control. Data analysis was performed with the ΔCt method and results were expressed as 2-ΔCt. Data are expressed as: mean ± SD, * *p* value ≤ 0.05; ** *p* value ≤ 0.01; **** *p* value ≤ 0.0001; number = not significant, test = unpaired *t*-test. (**D**) Graphic representation of SOD1-DT-regulated (black arrow) genes. Arrows show the sense or antisense transcription direction with the green and grey arrows indicating up- and non- regulated genes, respectively. Numbers indicate the distance in bp between SOD1-DT and its target genes.

**Figure 6 cells-12-02058-f006:**
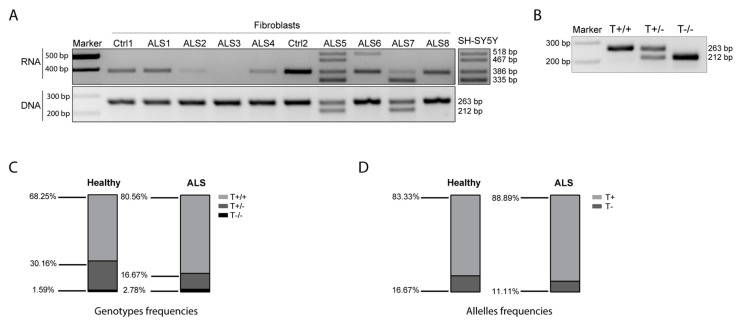
SOD1-DT in ALS. (**A**) RT-PCR analysis of SOD1-DT in skin fibroblasts derived from controls and ALS patients. (**B**) Representative RT-PCR showing T+/+, T+/− and T−/− genotypes. (**C**) T+/+, T+/− and T−/− genotypes frequencies in ALS patients’ analyzed gDNA samples. (**D**) T+ and T− alleles frequencies in ALS patients’ analyzed gDNA samples.

## Data Availability

All data generated or analyzed during this study are included in this article. Further inquiries can be directed to the senior corresponding author.

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
