# Peer review of "Characterization of SOD1-DT, a Divergent Long Non-Coding RNA in the Locus of the SOD1 Human Gene"

_cells, 2023, doi:10.3390/cells12162058_

Round 1

Reviewer 1 Report

Long noncoding RNA (lncRNA) is implicated in numerous biological functions and is associated with the nervous system. In this manuscript, Guerra et al. have characterized a lncRNA transcribed divergently from the SOD1 locus, namely SOD1-DT, in both SH-SY5Y cells and fibroblasts derived from ALS patients. They demonstrated that it has the potential to regulate the expression of local genes. Additionally, they characterized YY1 as a regulator of SOD1-DT. While SOD1-DT is an appealing molecule with potential implications for ALS, the current manuscript needs substantial revisions before it's ready for publication.

1, What is the length of this lncRNA? Is it a mRNA-like lncRNA that includes a polyA tail? Sequencing is not a precise method to determine the lncRNA sequence; it would be more appropriate to use the RACE technique to determine the start and end points of this lncRNA.

2, While qPCR is a feasible method to characterize lncRNA, uncontrolled amplification may occur during reverse transcription. Northern blot would be more suitable for directly illustrating the differential processing isoforms of SOD1-DT throughout the manuscript.

3, In Figure 2, the authors suggest that SOD1-DT is a nuclear lncRNA associated with chromatin. The direct visualization of RNA localization could be achieved more effectively by using RNA-FISH and co-staining with DAPI and histone markers. It will be interesting to show the sub-nuclear localization of this lncRNA, which may reflect its function.

4, In Figures 3A and 4B, given that total RNA could vary during differentiation and KD of SOD1-DT, experiments should use an equal cell number rather than equal RNA.

5, Has there been any observed impact on the differentiation of SH-SY5Y cells following the knockdown (KD) of SOD1-DT?

English is acceptable but some typos and errors should be revised. 

Reviewer 2 Report

Introduction flows a bit abruptly. Reordering the text where ALS is mentioned in more consistent manner could improve the readability of the text. 

Figure 3; it could have been better to see timeline from Day-0 to Day-6, rather than seeing two time points. 

In Figure 4; YY1 over expression leads to increased SOD1-DT, it would be nice to see that as a comparison point in Figure 5. This way, readers could compare low/control/high samples, and if TIAM1/HUNK/CCT8/FBXW11P1 changes in a similar manner, it would be stronger evidence. 

It is acceptable, more attention could be paid to flow of the text.

Round 2

Reviewer 1 Report

I would think the current version of the manuscript is at an acceptable level.